# Development of deep learning-assisted overscan decision algorithm in low-dose chest CT: Application to lung cancer screening in Korean National CT accreditation program

**Sihwan Kim**[1,2], **Woo Kyoung Jeong**[3], **Jin Hwa Choi**[4], **Jong Hyo Kim**[1,2,5,6,7,8], **Minsoo Chun**[8,9]*

**1** Department of Applied Bioengineering, Graduate School of Convergence Science and Technology, Seoul National University, Seoul, Republic of Korea, **2** ClariPi Research, Seoul, Republic of Korea, **3** Department of Radiology, Samsung Medical Center, Sungkyunkwan University School of Medicine, Seoul, Republic of Korea, **4** Department of Radiation Oncology, Chung-Ang University College of Medicine, Seoul, Republic of Korea, **5** Center for Medical-IT Convergence Technology Research, Advanced Institutes of Convergence Technology, Suwon, Republic of Korea, **6** Department of Radiology, Seoul National University College of Medicine, Seoul, Republic of Korea, **7** Department of Radiology, Seoul National University Hospital, Seoul, Republic of Korea, **8** Institute of Radiation Medicine, Seoul National University Medical Research Center, Seoul, Republic of Korea, **9** Department of Radiation Oncology, Chung-Ang University Gwang Myeong Hospital, Gyeonggi-do, Republic of Korea

* ms1236@caumc.or.kr

**Data Availability Statement:** All relevant data are within the manuscript and its Supporting Information files.

## Abstract

We propose a deep learning-assisted overscan decision algorithm in chest low-dose computed tomography (LDCT) applicable to the lung cancer screening. The algorithm reflects the radiologists' subjective evaluation criteria according to the Korea institute for accreditation of medical imaging (KIAMI) guidelines, where it judges whether a scan range is beyond landmarks' criterion. The algorithm consists of three stages: deep learning-based landmark segmentation, rule-based logical operations, and overscan determination. A total of 210 cases from a single institution (internal data) and 50 cases from 47 institutions (external data) were utilized for performance evaluation. Area under the receiver operating characteristic (AUROC), accuracy, sensitivity, specificity, and Cohen's kappa were used as evaluation metrics. Fisher's exact test was performed to present statistical significance for the overscan detectability, and univariate logistic regression analyses were performed for validation. Furthermore, an excessive effective dose was estimated by employing the amount of overscan and the absorbed dose to effective dose conversion factor. The algorithm presented AUROC values of 0.976 (95% confidence interval [CI]: 0.925–0.987) and 0.997 (95% CI: 0.800–0.999) for internal and external dataset, respectively. All metrics showed average performance scores greater than 90% in each evaluation dataset. The AI-assisted overscan decision and the radiologist's manual evaluation showed a statistically significance showing a *p*-value less than 0.001 in Fisher's exact test. In the logistic regression analysis, demographics (age and sex), data source, CT vendor, and slice thickness showed no statistical significance on the algorithm (each p-value > 0.05). Furthermore, the estimated excessive effective doses were 0.02 ± 0.01 mSv and 0.03 ± 0.05 mSv for each

**Funding:** This work was supported by the Korea Medical Device Development Fund grant funded by the Korean government (the Ministry of Science and ICT, the Ministry of Trade, Industry and Energy, the Ministry of Health & Welfare, the Ministry of Food and Drug Safety) (Project Numbers: 1711138600, 1711138601, KMDF_PR_20200901_0267). The funders had no role in study design, data collection and analysis, decision to publish, or preparation of the manuscript.

**Competing interests:** The authors have declared that no competing interests exist.

dataset, not a concern within slight deviations from an acceptable scan range. We hope that our proposed overscan decision algorithm enables the retrospective scan range monitoring in LDCT for lung cancer screening program, and follows an as low as reasonably achievable (ALARA) principle.

## Introduction

Although computed tomography (CT) technology has developed and achieved outstanding diagnostic accuracy, there remains concerns regarding radiation-induced cancers driving an as low as reasonably achievable (ALARA) movement [1–3]. The adoption of a low-dose CT (LDCT) protocol and the scan range optimization are exemplary efforts to lower imaging doses [4–10]. Particularly, LDCT has been popularly used in lung cancer screening program enabling to detect cancers in early stage, thereby reducing mortality rates [11, 12]. While the LDCT protocol has been adopted with various aspects, to the best of our knowledge, scan range selections still rely on a manual decision by the radiation technologist exhibiting intra- and inter-institution variations [13–19]. However, as a manual range selection is vulnerable to prevent excessive patients' doses, efforts should be made to provide optimal scan range and reduce inter-individual variability. Whereas excessive scan may increase unnecessary doses, the worse scenario is to scan with insufficient coverages requiring additional examination [20–22]. Both situations necessitate the scan range monitoring procedure either prospective or retrospective way.

According to the regulations for the operation of special medical equipment in South Korea, CT scanners have to be inspected every three years in terms of image quality and adequacy of the image acquisition method by an official CT certified agency such as the Korean institute for accreditation of medical imaging (KIAMI) [23]. Among various inspection items, an overscan audit is performed by an expert radiologist by visually inspecting scan ranges whether they are excessive or deficient to the criterion landmarks, which is laborious and subjective. Moreover, this manual auditing process could be performed for the only representative single scan, and the current process could not be applied to all patients' scan. Fully automated decision program might help to reduce subjectivity, be applied to all CT scans, and save time, human and cost resources. To evaluate the appropriateness of scan ranges, the objective criteria to determine overscan and underscan should be reasonably established according to the regions being scanned and clinical needs. In the audit process for lung cancer screening program in South Korea, the vocal cords and the kidney are used as landmarks for the superior and inferior limit, respectively.

Recent advancements in artificial intelligence (AI) technologies enabled to significantly reduce the time and cost resources of radiologists in various radiology applications, such as lesion classification, detection and segmentation [24–32]. Combining AI technology for organ segmentation and experts' decision rule-based logical operation, we developed automated overscan decision algorithm in lung cancer screening program and demonstrated the performances with internal and external dataset.

## Materials and methods

### Dataset

A total of 340 LDCT scans for lung cancer screening program was used for decision model development and validation, of which 290 scans were from our institution, named as internal

data, and 50 scans were collected from the 47 institutions, named as external data. While the internal data were approved by an institutional review board (IRB) at Seoul National University Hospital (IRB No. 2012-187-1186), no IRB approval was obtained for external data. A submission and the approval of the external data were subject to the National CT Accreditation Program conducted by the KIAMI. For both internal and external data, the informed consent was waived because all CT scans were retrospectively obtained, and all personal information tags in DICOM files were anonymized.

Among 290 internal data, 80 scans were used to develop landmark segmentation model as they have ground truth of landmark structures manually delineated by the expert radiologist with 21 years of experience. For details, 50 CT scans (Siemens 20 cases, GE 10 cases, Philips 20 cases) were used for model training and 10 CT scans (Siemens 4 cases, GE 2 cases, Philips 4 cases) were used for model tuning. The other 20 CT scans (Siemens 8 cases, GE 4 cases, Philips 8 cases) were used for its performance test. All internal data was reconstructed with a vendor-specific iterative reconstruction algorithm. Demographics and scan parameters for the internal data are described in Table 1. Note that the newly introduced tin filter technology was utilized in Siemens Force scanner [33–35].

**Table 1. Details on internal data.** Demographics and scan parameters are presented according to the data usage.

| Purpose | Vendor | Scanner name | # of scans (M/F) | Age | kVp | mAs | [a]AEC | Slice thickness (mm) | Reconstruction kernel | Note |
|---|---|---|---|---|---|---|---|---|---|---|
| Segmentation model development | Siemens | Definition Flash | 16 (16/0) | 69±9 | 120 | 30 | Off | 1 | B60f | Segmentation mask |
| | | Force | 16 (14/2) | 70 ±11 | Sn100[b] | 150 | Off | 1 | Br59d\3 | |
| | GE | Revolution | 16 (14/2) | 64 ±11 | 120 | 20±4 | On | 1.25 | Standard | |
| | Philips | Ingenuity | 16 (14/2) | 67 ±10 | 120 | 30 | Off | 1 | YC | |
| | | ICT | 16 (14/2) | 63 ±13 | 120 | 20 | Off | 1 | YC | |
| Decision model validation | Siemens | Definition flash | 30 (17/13) | 64 ±12 | 120 | 30 | Off | 1 | I70f\3 | Overscan tag |
| | | Force | 30 (17/13) | 61 ±13 | Sn100 | 150 | Off | 1 | Br59d\3 | |
| | GE | Revolution | 30 (21/9) | 62 ±13 | 120 | 23±3 | On | 1.25 | LUNG | |
| | | Discovery | 30 (15/15) | 61 ±13 | 120 | 42 | Off | 1.25 | LUNG | |
| | Philips | Ingenuity | 30 (16/14) | 64 ±14 | 120 | 30 | Off | 1 | YC | |
| | | IQon | 30 (18/12) | 65 ±11 | 120 | 20 | Off | 1 | YC | |
| | Canon | Aquilion | 30 (15/15) | 71 ±19 | 120 | 97 ±26 / 30 | On: 8 / Off: 22 | 1 | FC15-H | |

[a]AEC: Automatic exposure control,

[b]Sn100: 100 kVp with tin filter attached.

The other 210 internal data and 50 external data were used to demonstrate the performance of the overscan decision model. Fifty external data were obtained from 20 university hospitals and 27 private hospitals, and were consisted of 26 different CT scanners with various low-dose CT scan conditions. Due to their huge diversity, detailed information is presented in S1 Table. All 260 data were marked with overscan tags established by the radiologist's evaluation. Data from internal and external sources with overscan tagging showed rates of 22.4% and 32%, respectively.

## Overscan decision criteria

We applied the KIAMI's overscan decision criteria in lung cancer screening CT images in the algorithm development [36]. In these criteria, the upper end of the vocal cords and the lower end of the kidney were used as the reference landmarks to represent a superior and inferior scan limit, respectively. However, vocal cords exhibit an irregular anatomical shapes and ambiguity in exact localization even by human observers (Fig 1A). Rather, we determined a thyroid cartilage as a substitute to the vocal cord (Fig 1B) because a thyroid cartilage is easier to be segmented presenting high contrast in CT images and covers the vocal cords in the longitudinal direction [37]. A kidney was used as an inferior-direction landmark (Fig 1C).

## Overscan decision algorithm

The development workflow consisted of three major stages: a deep learning-based fully automated landmark segmentation stages (thyroid cartilage or kidney), the rule-based logical operations for the landmark localization, and the final determination of the overscan direction and its length (Fig 2).

**AI based landmark segmentation.** In the first stage, deep-learning-based fully automated segmentation was implemented using a 2D-image based U-Net model [38]. Key imaging features of the target object were automatically extracted by using concatenated encoding-decoding architecture. The model was trained by using a supervised learning with manually labelled landmarks. For efficient parameter learning with relatively small data, they were randomly augmented by rotation (within 5 degrees) and translation (within 5 pixels) in each training iteration. In addition, the binary cross-entropy loss for two classes (one for the background and the other for the target object) was optimized with an iterative learning process, and an optimal training iteration was determined by an early stopping method using the AI-model

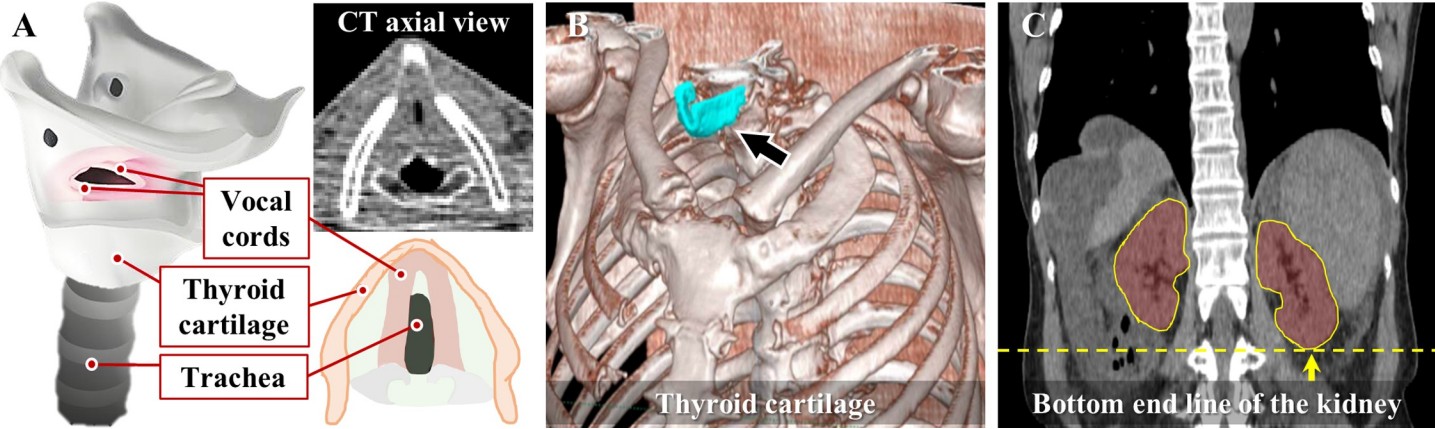

**Fig 1. Representative landmarks in overscan decision.** (A) Vocal cord as an initial superior-side landmark, (B) Thyroid cartilage as a replaced superior-side landmark, (C) Kidney as an inferior-side overscan landmark.

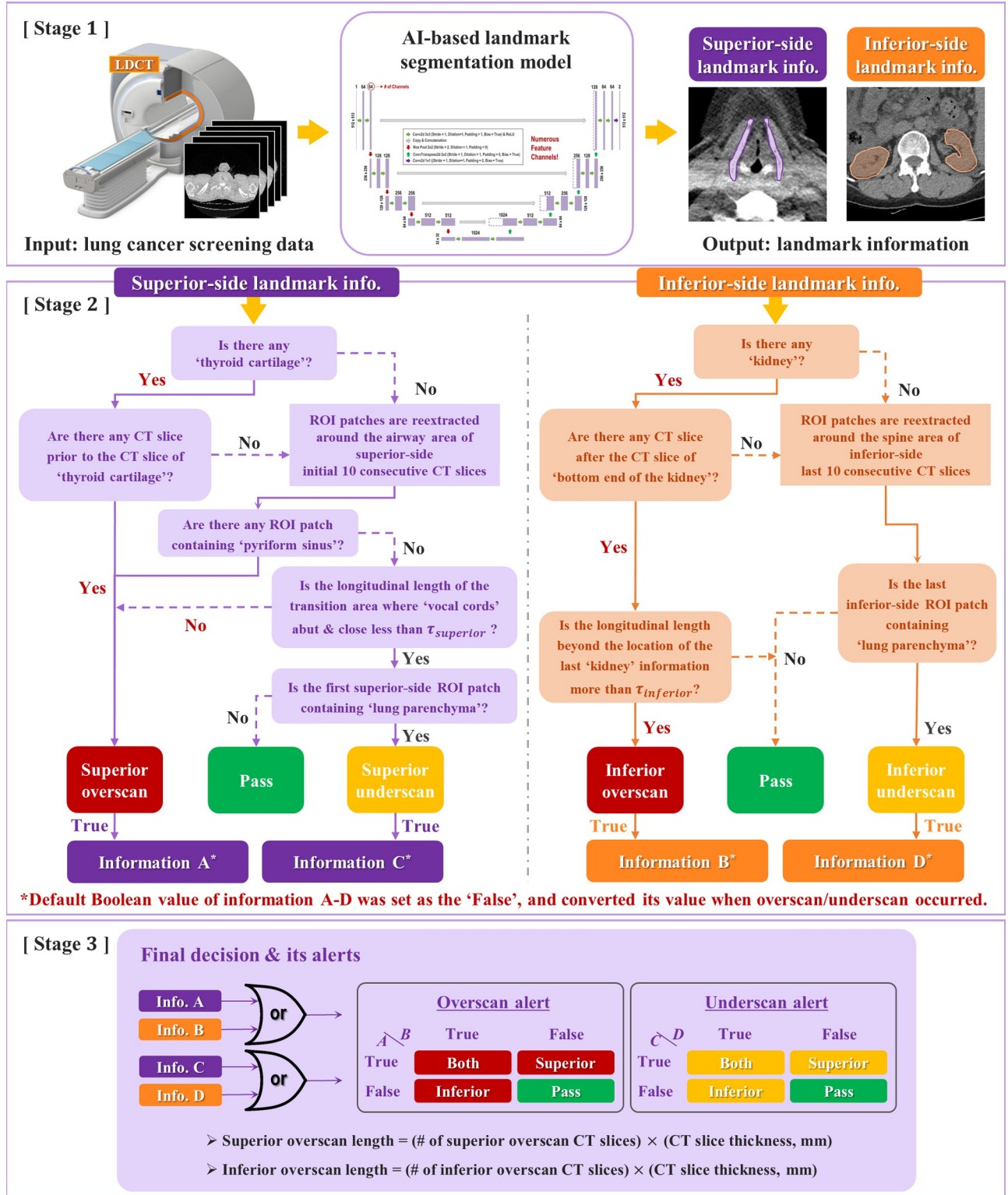

**Fig 2. Schematic diagram of overscan determination process.** Stage 1: AI-assisted landmark segmentation. Stage 2: Rule-based logical operations. Stage 3: Final overscan decision and its alerts.

tuning dataset [39]. To maximize the performance, two separate U-Net based segmentation models were trained for each landmark. When the lung cancer screening data were imported to the model, landmark's binary masks were generated and passed to the rule-based logical operation stage.

**Sub-algorithm for rule-based logical operation.** In the second stage, there are two determination branches according to the landmark types. Each branch comprised of rule-based logical operations, reflecting the human decision-making process. The identification information (information A-D in Fig 2) of overscan/underscan was logically acquired for each determination branch, based on the information of predefined landmark presence and its relative position to initial/last CT slice.

When the thyroid cartilage was localized, the existence of images prior to the thyroid cartilage was examined to decide the superior-side overscan. If the thyroid cartilage was not detected, ROI patches were scanned for the ten-consecutive superior slices around the airway region to cope with false-negative cases of the landmark. The ROI patches had $64 \times 64$ pixels and the airway region was localized by thresholding CT numbers from -1000 HU to -900 HU. The algorithm decided the overscan when the ROI patches detected the pyriform sinus. In case of no pyriform sinus in the patch, the length of vocal cords being touched each other or closed are examined. While the overscan decision was made when the length exceeded the threshold ($\tau_{superior}$), the algorithm further examined whether the patches contained the lung parenchymal tissue if the length was less than the threshold. The decision was made as appropriate scan range or underscan according to the existence of the lung parenchymal tissue within the patches. The optimal $\tau_{superior}$ was empirically obtained by using the internal data.

Similarly, in inferior-side determination, the inferior-side overscan was decided with relative location of the landmark organ, kidney. To cope with a real absence of the kidney or prevent false-negative cases, ROI patches with a $64 \times 64$-pixel area were extracted around the spine region within the ten-consecutive inferior-side CT slices. The spine region was detected by thresholding CT numbers of 150 HU. If the range of CT scanning exceeded the location of the last kidney, an inferior overscan was determined accordingly. An optimal inferior-side threshold value ($\tau_{inferior}$) for the exceeding length of the inferior-side scan was determined by referencing a possible movement range of the kidney by intrinsic human motion (e.g., breathing or peristalsis of digestive organs). As in the case of the superior underscan, the information on lung parenchyma presence was used to filter out the inferior underscan. If the last inferior-side ROI patch included the region of the lung parenchyma, it was concluded as the inferior underscan. Scan range identification information (Fig 2, information A-D), as the results for logical operation stage, were set as 'False' at default. When the overscan or underscan occurred, the Boolean value was converted from 'False' to 'True'. The final Boolean values in information A-D were transmitted to the last decision stage.

**Final decision and its alerts.** In the last stage, the final determination of the overscan direction and its length was made through the identification information resulted from rule-based logical operations for both superior-side and inferior-side. When the identification information on the overscan (Fig 2, information A or information B) and the underscan (Fig 2, information C or information D) were given as the Boolean-type data (e.g. 'True' or 'False'), the overscan/underscan direction was determined by the logical OR gate. After finishing each detection process, the algorithm alerted scan range status. While the underscan detection is limited to alerting the scan status, overscan detection further includes the overscan length calculation. The overscan length was calculated by multiplying the slice thickness of the reconstructed CT image and the number of slices determined as the overscan.

## Evaluation design

**Segmentation model evaluation.** The segmentation performance was evaluated with dice similarity coefficient (DSC) for each patient [40]. The DSC ranged from 0 to 1 corresponding to no-overlap and complete overlap, respectively. The evaluation was with AI-model tuning and test data of the development dataset for thyroid cartilage and kidney, respectively. Mean and standard deviation of DSC for those data were calculated.

**Optimal threshold cut-off.** The optimal $\tau_{superior}$ was heuristically determined as 9 mm according to the receiver operating characteristic (ROC) analysis with internal data. In contrast, the $\tau_{inferior}$ was determined as 3 mm by considering the intrinsic movement range of the kidney [41].

**Metrics for decision model validation.** The area under the receiver operating characteristic (AUROC) metric was used to assess the overall algorithm performance with the linear approximation of ROC curves. At the given optimal $\tau_{superior}$ and $\tau_{inferior}$, the accuracy, sensitivity, and specificity were calculated by using the standard logit method, and their corresponding 95% confidence intervals (CI) were calculated with the Clopper-Pearson's method [42–45]. The Cohen's kappa was calculated by using McHugh's formula [46]. All metrics ranged from 0 to 1 and are expressed as percentages except for kappa and AUROC. Fisher's exact test was used to test whether the developed algorithm could distinguish overscan cases [47]. A significance level was set at 0.05. All performance metrics were calculated by using MedCalc software (Version 20.023, MedCalc Software Ltd., Mariakerke, Belgium).

**Evaluation procedure.** First, the AUROC was calculated using each decision model validation data, and the optimal threshold was selected as a cut-off value under the best algorithm performance while maintaining conservative determination criteria (e.g., the shorter the CT scan range became, the lower the radiation exposure that could be achieved.) through ROC analysis using the internal data. Second, based on the selected cut-off value, as evaluation metrics, accuracy, sensitivity, and specificity were calculated on the validation data. Finally, a univariate logistic regression analysis was used to assess the generalizability of the developed algorithm, as well as its accuracy on each independent variable [48].

**Univariate logistic regression analysis.** The univariate logistic regression was performed to demonstrate statistical significances of the algorithm according to the patient's characteristics (e.g. age and sex), data source, CT vendor, and slice thickness. The analysis was conducted based on the confusion matrix of rearranged data with respect to the five independent variables using both internal and external data (S2 Table). The dependent variable was the correctness of the overscan decision. It evaluated the accuracy with respect to the components of each variable and calculated their odds ratios (ORs) with 95% CIs. The IBM SPSS Statistics software (version 25.0; IBM Corp., Armonk, NY, USA) was used, and the statistical significance was set as $p$-value $< 0.05$.

**Excessive effective dose estimation for overscan.** The excessive effective dose was estimated by multiplying volume CT dose index ($CTDI_{vol}$), overscan length and the tissue conversion factor ($k$). The $CTDI_{vol}$ was obtained from either CT DICOM header or a structured dose report. The $k$ value was referenced from the report of American Association of Physicists in Medicine (AAPM) and was 0.0059 and 0.015 for the superior and inferior-side overscan, respectively, which corresponds to neck and abdomen region [49]. The detailed scan parameters and the excessive effective dose was provided in S3 Table.

# Results

## Segmentation model evaluation

The DSCs of landmark segmentations on LDCT AI-model tuning data were 0.76 ± 0.09 and 0.88 ± 0.14 for thyroid cartilage and kidney, respectively. On LDCT testing data, the DSCs for thyroid cartilage and kidney were 0.79 ± 0.25 and 0.93 ± 0.09, respectively. Except for slices of the superior-side overscan (Fig 3A), the AI-assisted segmentation showed DSCs over 90% with manual delineation within a general scan range (Fig 3B and 3C). In the superior-side scanning, 90% of the overscan data were scanned up to position of the pyriform sinus level.

## Decision model validation

Fig 4 shows the ROC curves of the overscan decision algorithm. The AUROC values were 0.976 (95% CI: 0.925–0.987) and 0.997 (95% CI: 0.800–0.999) for the internal and external data, respectively. For the optimal $\tau_{superior}$ and $\tau_{inferior}$, calculating from 2x2 contingency table (Table 2), all evaluation metrics showed greater than 90%. In decision model validation, the accuracy of the algorithm was 96.67 (95% CI: 93.25%-98.65%) and 96.00 (95% CI: 86.29%-99.51%) for internal and external data, respectively. While the sensitivity was shown as 97.87% (95% CI: 88.71%-99.95%) and 93.75% (95% CI: 69.77%-99.84%), the specificities were 96.32% (95% CI: 92.16%-98.64%) and 97.06% (95% CI: 84.67%-99.93%) for internal and external data, respectively. The Fisher's exact test between our decision algorithm and the experienced radiologist showed statistical significances ($p < 0.001$).

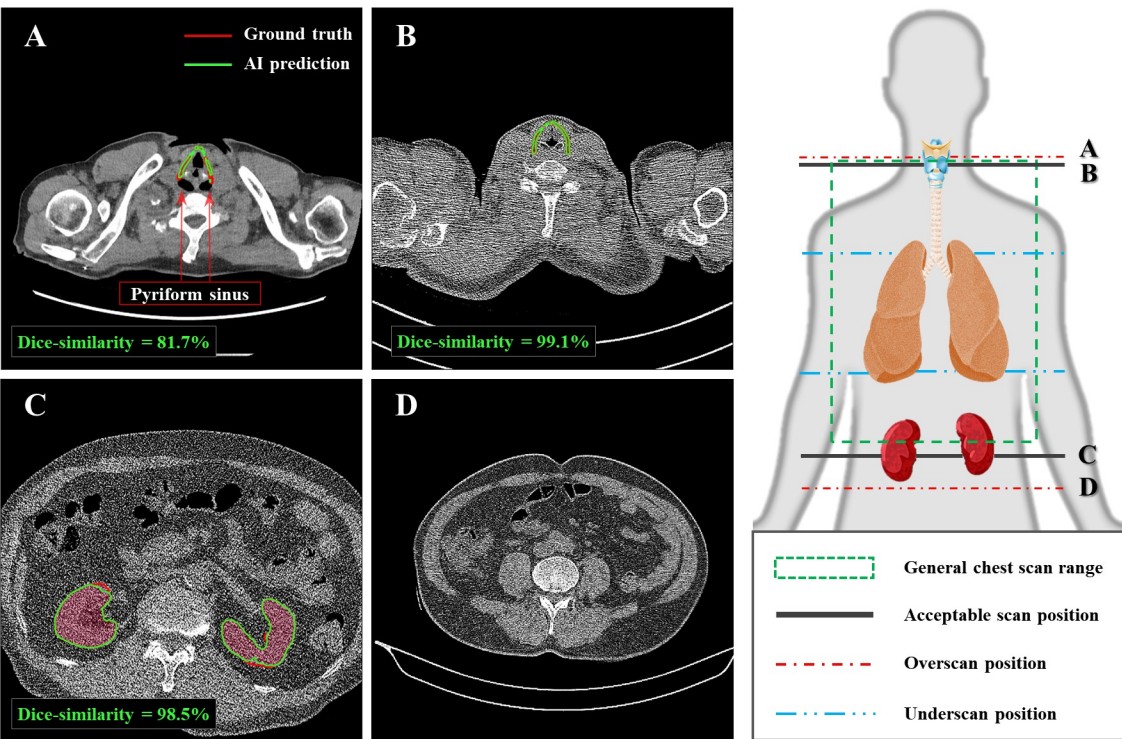

**Fig 3. Sample landmark segmentation comparison between AI-assisted model and the experienced radiologist.** The results were visualized at different scanning position; (A) segmentation result of superior-side scan at overscan position, (B) that of superior-side scan at acceptable scan position, (C) that of inferior-side scan at acceptable scan position, and (D) inferior-side scanning example at overscan position. Ground truth label and AI-model's prediction result were displayed with red and green lines, respectively. (W/L: 400/40).

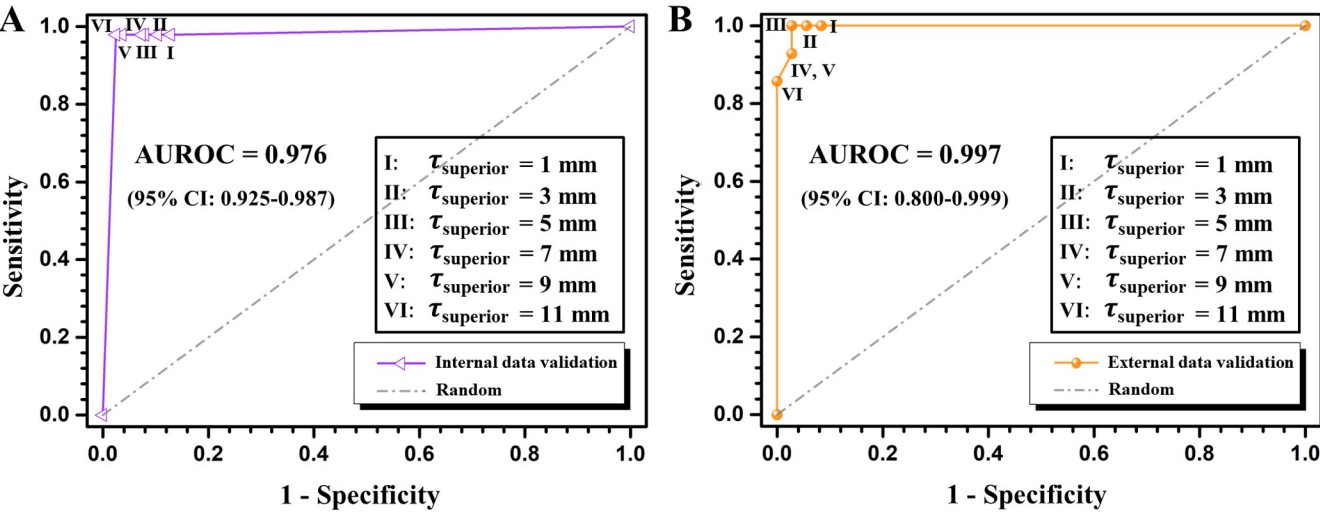

**Fig 4.** ROC curve and AUROC values for internal (A) and external data (B), respectively.

## Univariate logistic regression analysis

Table 3 shows the results of univariate logistic regression analysis. The univariate analysis revealed that age, sex, data source, CT vendor, and slice thickness had no statistically significant influence on the algorithm's decision. The individual accuracy of the algorithm for the rearranged data according to the five independent variables was above 95%.

## Excessive effective dose estimation for overscan

Detailed scan parameters and estimated excessive effective doses for overscan data were provided in S3 Table. The mean $CTDI_{vol}$ of overscan cases was 2.42 ± 1.69 mGy and 2.54 ± 0.65 mGy, and the mean overscan length was 1.59 ± 0.7 cm and was 1.66 ± 1.41 cm for internal and external data, respectively. The estimated effective doses caused by the overscan increased about 0.02 ± 0.01 mSv and 0.03 ± 0.05 mSv for internal and external data, respectively.

## Discussion

The purpose of our study was to develop a deep learning-assisted algorithm to discriminate the overscan cases of lung cancer screening program in LDCT scan. Most of the previous studies on overscan determination [10, 18] in chest CT scans mainly focused on scan range

**Table 2. Contingency table of decision model validation with internal and external data.**

| Contingency table of internal data | | Gold standard by an audit-experienced radiologist | |
|---|---|---|---|
| | | Overscan | None |
| Algorithm prediction | Overscan | 46 | 6 |
| | None | 1 | 157 |
| Contingency table of external data | | Gold standard by an audit-experienced radiologist | |
| | | Overscan | None |
| Algorithm prediction | Overscan | 15 | 1 |
| | None | 1 | 33 |

**Table 3. Univariate logistic regression analysis.** The internal and external data were rearranged to satisfy pre-defined univariate condition for each variable.

| Independent variable | | Univariate analysis | | | Performance | |
|---|---|---|---|---|---|---|
| | | OR[a] | 95% CI[b] | p-value | Accuracy (%) | 95% CI |
| Age (n = 210) | age ≤ 64 yr (n = 106) | 1 | - | - | 98.1 | [93.4, 99.8] |
| | age > 64 yr (n = 104) | 0.381 | [0.07, 2.01] | 0.255 | 95.2 | [89.1, 98.4] |
| Sex (n = 210) | Male (n = 119) | 1 | - | - | 95.8 | [90.5, 98.6] |
| | Female (n = 91) | 1.952 | [0.37, 10.30] | 0.431 | 97.8 | [92.3, 99.7] |
| Data source (n = 260) | Internal (n = 210) | 1 | - | - | 96.7 | [93.3, 98.7] |
| | External (n = 50) | 0.828 | [0.17, 4.11] | 0.817 | 96.0 | [86.3, 99.5] |
| CT vendor (n = 260) | Siemens (n = 83) | 1 | - | - | 96.4 | [89.8, 99.3] |
| | GE (n = 71) | 0.85 | [0.17, 4.35] | 0.845 | 95.8 | [88.1, 99.1] |
| | Philips (n = 65) | 1.181 | [0.19, 7.29] | 0.858 | 96.9 | [89.3, 99.6] |
| | Canon (n = 41) | 1.5 | [0.15, 14.88] | 0.729 | 97.6 | [87.1, 99.9] |
| Slice thickness (mm, n = 260) | 1 ≤ t < 2 (n = 238) | 1 | - | - | 96.6 | [93.5, 98.5] |
| | 2 ≤ t ≤ 5 (n = 22) | 0.73 | [0.09, 6.12] | 0.772 | 95.5 | [77.2, 99.9] |

[a]OR: Odds ratio,

[b]CI: Confidence interval

delimitation in topograms, but not in axial CT slices. Automatic scan range delimitation in the topogram domain would be a meaningful approach because the topogram defined the regions being scanned prior to actual scans. In these regards, scan range recommendations in topogram stage enable to reduce the workload and inter-operator variability, and there have been various related researches. However, it is challenging and premature because there are opaque shadings and intensity variation across CT vendors in topograms, and even AI-assisted approaches are still hard to localize key landmarks [22]. Furthermore, localization comparison between scout and axial images should be preceded for their clinical routine [50]. Although optimal scan range recommendations prior to CT scans is meaningful, we mainly focused to establish more objective system to retrospectively assess the overscan length and its corresponding excessive doses, and there are no previous related studies.

Our proposed algorithm presented remarkable performances showing values greater than 95% and 97% of accuracy and AUROC, respectively. The developed algorithm showed almost perfect agreement with experienced radiologists, with kappa values of 0.907 and 0.908 for internal and external dataset, respectively. Furthermore, the Fisher's exact test demonstrated that the proposed algorithm had an ability to detect the overscan similar to the audit-experienced experts. The logistic regression analysis showed no statistical significances with various potential variables in LDCT, such as age, sex, data source, CT vendor, and slice thickness. These results suggest that the algorithm has a decision-making ability similar to that of a radiologist, and it has a high potential for generalizability in LDCT. In other words, despite differences in LDCT protocols between hospitals, the proposed algorithm could be generalized for the majority of patients taking lung cancer screening in LDCT, and this can be applicable to hospitals' independent quality control practice.

Considering the mean effective dose being about 0.02 mSv in case of plain PA chest radiograph, the estimated excessive effective dose in our study are equivalent or 1.5 times greater than those in chest PA on average [51]. In the worst overscan case of the study, the excessive effective dose caused by overscan could reach up to 11 times of those by plain PA chest radiograph. However, all those effective dose levels were much below the possible chromosomal damage level (5 mSv) by the X-ray radiation [52]. In the case of slight deviations from the

acceptable scan range, especially for LDCT scan, it is thought that excessive effective dose is not a concern.

This study had a few limitations. First, a subjective bias could affect the gold standard, even though it was created by an experienced radiologist. In the future study, it might better to secure an objectivity by utilizing records of more than two experts. Second, in the AI model segmentation stage, there might exist segmentation errors for both superior and inferior direction (S1 Fig). The main reasons were extreme quantum-noise of LDCT and lack of a diversity of trained data distribution. However, despite the segmentation errors, the developed algorithm minimized its vulnerability to error cases by aggregating the 2D segmented landmark information into 3D volumetric information. Furthermore, several segmentation errors could be filtered out and improved by the logical operation of the second-stage. Third, analyzing the contingency table in both internal and external data, the algorithm still had a few false-negatives (algorithm = none, radiologist = overscan) and false-positives (algorithm = overscan, radiologist = none). The reason for false-negative occurrences in both data was that the length of the transition area in superior-side scan was not over the pre-defined overscan cut-off threshold. In most of the false-positive cases, the partial segmentation failure of the landmark at the initial or last slice position was considered as a main reason in our algorithm. Although our algorithm could coordinate false-positives and false-negatives by adjusting the cut-off threshold ($\tau_{superior}$ or $\tau_{inferior}$), the optimal threshold might require a large scale and multi-centre investigation. Fourth, we could not prospectively suggest the optimal scan range prior to CT scan while other topogram-based approaches did. As stated above, the further thorough demonstration of its clinical appropriateness between topogram and axial slices retrospectively with large amount of dataset enable to reliably suggest the optimal scan range, thereby achieve fully automated scan range suggestion practice in lung cancer screening. Lastly, there exists only few underscan cases (0.4%), and the separate study to evaluate the underscan is necessary as a future research topic. Also, the demonstrations were limited to LDCT, it could be applied to standard-dose CT as applications work better in standard dose CT than LDCT.

Beyond limitations, by using the developed algorithm, the radiologists only need to pay attention to the alerting cases by the overscan/underscan. The less overscan rate, the less radiologist's workload could be accomplished. In comparison with every single patient audit procedure, as the overscan occurred in internal and external datasets at a frequency of 22.4% and 32%, it was estimated that the developed algorithm could reduce the workload of overscan range check by 68% and 77.6% for each dataset. We also expect that combinations of our algorithm with radiation dose and image quality monitoring could establish the fully automated and integrated CT quality system for every patient [53–57]. By reducing human and time resources with full automation, it is available to equip high throughput and objective quality monitoring platform for the entire CT scans.

## Conclusions

We developed the deep learning-assisted overscan decision algorithm including three stages of AI-based landmark segmentation model, rule-based logical operation, and final determination. In the demonstration with 210 and 50 lung cancer screening cases for internal and external dataset, the algorithm showed values of greater than 96.0% and 97.6% of accuracy and AUROC, respectively. For LDCT chest screening, the excessive effective dose is not a concern within the slight deviations from the acceptable scan range. We hope that the combination of the proposed algorithm along with multi-parametric image quality assessment, and radiation dose monitoring program will lead the scan range and protocol optimization, and contribute patients' radiation safety by following ALARA principle. Furthermore, hospitals enable to

establish an independent quality monitoring platform and this automated system will allow high throughput and objective quality control to improve the entire CT practice.

## Supporting information

**S1 Table. Detailed scan parameters for external data.**
(DOCX)

**S2 Table. Confusion matrix of each variable in univariate logistic analysis.** The variable consists of age, sex, data source, slices thickness and CT vendors. "Correct" and "Incorrect" indicate the success and fail of overscan detection, respectively.
(DOCX)

**S3 Table. Excessive effective doses caused by overscans for internal and external dataset.**
(DOCX)

**S1 Fig. Error examples for AI-assisted segmentation model.** Segmentation results for superior-side (A-C) and inferior-side (D-F) were visualized in 2D axial view and 3D rendering view. The use of only two-dimensional information on landmarks' existence may exhibit an entire (B) or partial (E) segmentation failure, while the use of volumetric information stacked from 2D information has been improved in landmark detections (C, F). Ground truth label and AI predictions were marked as red and green lines, respectively. (W/L = 400/40 HU).
(TIF)

## Acknowledgments

This study was performed with the help of the Korean Institute for Accreditation of Medical Imaging.

## Author Contributions

**Conceptualization:** Jong Hyo Kim, Minsoo Chun.

**Data curation:** Jin Hwa Choi.

**Formal analysis:** Sihwan Kim.

**Funding acquisition:** Minsoo Chun.

**Investigation:** Sihwan Kim.

**Methodology:** Sihwan Kim.

**Project administration:** Minsoo Chun.

**Resources:** Woo Kyoung Jeong, Minsoo Chun.

**Software:** Sihwan Kim.

**Supervision:** Woo Kyoung Jeong, Jong Hyo Kim, Minsoo Chun.

**Validation:** Woo Kyoung Jeong, Jin Hwa Choi.

**Visualization:** Sihwan Kim.

**Writing – original draft:** Sihwan Kim.

**Writing – review & editing:** Jong Hyo Kim, Minsoo Chun.

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
