## [Decision Letter · Decision Letter 0]

20 Jun 2022

PONE-D-22-05608Deep Learning-Assisted Algorithm for Automated Overscan Determination in Low-dose Chest CT: A Nationwide Study on Korean CT Accreditation ProgramPLOS ONE

Dear Dr. Chun,

Thank you for submitting your manuscript to PLOS ONE. After careful consideration, we feel that it has merit but does not fully meet PLOS ONE’s publication criteria as it currently stands. Therefore, we invite you to submit a revised version of the manuscript that addresses the points raised during the review process. Please pay particular attention to and explain the highly asymmetric confidence intervals in Results.

We look forward to receiving your revised manuscript.

Kind regards,

Mingwu Jin, Ph.D.

Academic Editor

PLOS ONE

**Journal requirements:**

“This work was supported by the Korea Medical Device Development Fund grant funded by the Korean government (the Ministry of Science and ICT, the Ministry of Trade, Industry and Energy, the Ministry of Health & Welfare, the Ministry of Food and Drug Safety) (Project Number: 1711138600, 1711138601, KMDF_PR_20200901_0267).”

Please include this amended Role of Funder statement in your cover letter; we will change the online submission form on your behalf

4. Thank you for stating the following in the Funding Section of your manuscript:

“This work was supported by the Korea Medical Device Development Fund grant funded by the Korean government (the Ministry of Science and ICT, the Ministry of Trade, Industry and Energy, the Ministry of Health & Welfare, the Ministry of Food and Drug Safety) (Project Number: 1711138600, 1711138601, KMDF_PR_20200901_0267).”

“This work was supported by the Korea Medical Device Development Fund grant funded by the Korean government (the Ministry of Science and ICT, the Ministry of Trade, Industry and Energy, the Ministry of Health & Welfare, the Ministry of Food and Drug Safety) (Project Number: 1711138600, 1711138601, KMDF_PR_20200901_0267).”

Reviewers' comments:

Reviewer's Responses to Questions

**Comments to the Author**

1. Is the manuscript technically sound, and do the data support the conclusions?

Reviewer #1: Partly

Reviewer #2: Yes

2. Has the statistical analysis been performed appropriately and rigorously? 

Reviewer #1: No

Reviewer #2: Yes

3. Have the authors made all data underlying the findings in their manuscript fully available?

Reviewer #1: No

Reviewer #2: Yes

4. Is the manuscript presented in an intelligible fashion and written in standard English?

Reviewer #1: Yes

Reviewer #2: Yes

5. Review Comments to the Author

Reviewer #1: This paper proposed an AI-based target organ segmentation (plus logical operation) method to automatically determine the overscan of LDCT of chest screening. The authors used a relatively small data set (without details on the patient cohort information, age, size, only mentioned sex in the results). My main concern is on the handling of the under scan. As the authors stated in the introduction, under scan actually has more negative dose concerns than the overscan if a new scan has to be repeated. But in the current study, underscdan is also considered as “pass”. I understand that for underscan you might not be able to detect the target organ slices, or only part of them, using the AI. But equal effort should be spent on the under scan if the purpose is to avoid extra dose.

Following the above comment, would suggest the authors report on the dose increases in real values for over scan/under scans they detected. This will put the value of this study into content. How much of dose increases are we detecting?

Another follow up question is that it is not clear to me exactly how the target organ information is used after detection, if there’s space, more figures/graphs showing the exact boundary of the organs in regard to a specific scan range will be very helpful.

A few minor comments:

Line 67-69, redundant statement as Line 55-57. Only need to keep one place.

Line 257-258, it seems from Table 3 that the p values are the numbers determined, not a range, should be something like, p = 0.431, not p<0.431. Please double check.

For all the statistical results with CI, not sure why quite some values are very skewed, i.e. not close to the center of the CI. For example, Table 3, Odds ratio, typically should be symmetric for the estimated value within the CI. And Table 2, sensitivity for external data, with such a large CI, 69% to 99%, the estimated value is still 93%? Can you double check?

S1 Table, for Philips scanner, iDose is the iterative recon strength level, not the recon kernel. Same problem for S2 Table, ASiR-V is the algorithm, not the kernel. On the opposite side, S3 Table 3, all the recon kernels were correctly reported, but under the “reconstruction method” column. Please correct those.

Reviewer #2: Overall review: The novelty of the manuscript is the determination of overscan using axial slices. The U-net-based segmentation is not novel but is suitable for the work. It is made clear by the authors that this automated tool is developed for the purpose of retrospectively monitoring scan range selection in clinical practices, and it cannot be used to set the scan range for the patient before the scan.

Major points:

1. As the ultimate goal is to reduce radiation exposure for the benefit of the patients, I'd like to know the reasons why the authors chose to focus on monitoring the scan range retrospectively instead of predicting the appropriate scan range before the scan.

2. Theoretically, the method of deep learning segmentation and using landmarks to identify scan range can be applied to tomograms as well. If topograms were used, they could potentially be used to prevent overscan. What is the main reason to use axial slices over topograms besides the lack of images?

3. When presenting the segmentation results, please provide example images segmented by the DL algorithm and the radiologist.

4. Please provide the confusion table before you calculate the accuracy, sensitivity, specificity, PPV, and NPV in Table 2, since all those metrics came from the same confusion table. Please provide the contingency table for your Fisher's exact test as well. Similarly, please provide the contingency tables for your univariate OR analysis as an addition to Table 3.

5. It is good to see the univariate analysis in Table 3. But it is also important to know when your automated tool fails, especially when a deep learning algorithm is involved. Therefore, I'd love to see an error analysis. You have acknowledged that segmentation may affect it. If so, please add a couple of image examples along with the dice scores to Discussion. If the segmentation performance is correlated with any of the variables listed in Table 3, please add the dice scores to Table 3 as well.

6. The figure and table captions are not self-explanatory. Please explain what's in the figure/table in the caption with more details.

Minor points:

1. Typo on line 242: The accuracy of internal data ranges from 93.24% while the number is 93.25% in Table 2.

6. PLOS authors have the option to publish the peer review history of their article (what does this mean?). If published, this will include your full peer review and any attached files.

Reviewer #1: No

Reviewer #2: No

---

## [Author Response · Author response to Decision Letter 0]

28 Jul 2022

All replies are wihtin "Response to reviewer" file.

---

## [Decision Letter · Decision Letter 1]

9 Sep 2022

PONE-D-22-05608R1Development of Deep Learning-Assisted Overscan Decision Algorithm in Low-dose Chest CT: Application to Lung Cancer Screening in Korean National CT Accreditation ProgramPLOS ONE

Dear Dr. Chun,

Thank you for submitting your manuscript to PLOS ONE. After careful consideration, we feel that it has merit but does not fully meet PLOS ONE’s publication criteria as it currently stands. Therefore, we invite you to submit a revised version of the manuscript that addresses the points raised during the review process.

We look forward to receiving your revised manuscript.

Kind regards,

Mingwu Jin, Ph.D.

Academic Editor

PLOS ONE

Journal Requirements:

Reviewers' comments:

Reviewer's Responses to Questions

**Comments to the Author**

1. If the authors have adequately addressed your comments raised in a previous round of review and you feel that this manuscript is now acceptable for publication, you may indicate that here to bypass the “Comments to the Author” section, enter your conflict of interest statement in the “Confidential to Editor” section, and submit your "Accept" recommendation.

Reviewer #1: All comments have been addressed

Reviewer #2: (No Response)

2. Is the manuscript technically sound, and do the data support the conclusions?

Reviewer #1: Yes

Reviewer #2: No

3. Has the statistical analysis been performed appropriately and rigorously? 

Reviewer #1: I Don't Know

Reviewer #2: Yes

4. Have the authors made all data underlying the findings in their manuscript fully available?

Reviewer #1: Yes

Reviewer #2: Yes

5. Is the manuscript presented in an intelligible fashion and written in standard English?

Reviewer #1: Yes

Reviewer #2: Yes

6. Review Comments to the Author

Reviewer #1: I want to thank the authors for their effort on the revised manuscript. Most of my previous comments are addressed.

Just have two more comments related to the added/revised components:

1. Fig. 3, very nice visualization. Just why there are two acceptable locations for the inferior side (C and D)? Need a little more explanation in the text. I saw some descriptions on the supplemental materials about the AI failing in 2D. But still not clear why there are two acceptable positions shown in Fig. 3. Also do you have any over scan images for the inferior side?

2. The overall additional dose from overscan. As I have expected, the mean added effective dose due to overscan is really low. 0.02 mSv should not be any concern when the opposite side is underscan with missed information. I see that the authors tried to defend that the dose is not low in order to justify the value of the study. I kind of disagree with that logic. It is a good news that after this study, we found out that the dose from overscan is really low, not to any concern. The study is still valuable. It is not necessary to overstate the dose issue to justify the study.

Reviewer #2: In this revision, the authors clarified that the purpose of the work is to reduce subjectivity of auditing of overscans (by radiologists) and save time/cost/resources. However,

1) The gold standard used in this study was from only 1 radiologist.

2) If the authors wish to reduce the radiologists' burden to audit overscans, it is probably better to completely eliminate the false negatives (algorithm = None, radiologist = Overscan) so that the radiologists do not need to review every single non-overscan cases to confirm that 1 false negative (according to Table 2, there is 1 with internal data and 1 with external data). Therefore, I suggest the authors do a more in-depth analysis on why the false negatives occurred and maybe also tune your network to prioritize the elimination of false negatives even if it means the false positive cases (algorithm = Overscan, radiologist = None) may increase greatly. Note that this comment is based on my assumption that overscans are rare compared to appropriate scan range and that no overscan case can be missed (therefore the review by radiologist).

3) I assume that the every overscan case decided by the algorithm would be reviewed by radiologists so that no operator would be wrongfully accused. I am not sure how often/rare overscan occurs in routine LDCT lung scans, but the authors can probably provide an estimate on how much time it could save by only having to review the overscans brought up by the algorithm.

I believe the algorithm proposed by the authors is potentially useful. However, I do not think the authors have explained how to realistically help reduce the burden of auditing overscans on radiologists. Just having an algorithm that has good overall accuracy is not enough because radiologists would still need to review all the cases.

7. PLOS authors have the option to publish the peer review history of their article (what does this mean?). If published, this will include your full peer review and any attached files.

Reviewer #1: No

Reviewer #2: No

---

## [Author Response · Author response to Decision Letter 1]

12 Sep 2022

All responses to review comments are attached as a separate file.

---

## [Decision Letter · Decision Letter 2]

19 Sep 2022

Development of Deep Learning-Assisted Overscan Decision Algorithm in Low-dose Chest CT: Application to Lung Cancer Screening in Korean National CT Accreditation Program

PONE-D-22-05608R2

Dear Dr. Chun,

We’re pleased to inform you that your manuscript has been judged scientifically suitable for publication and will be formally accepted for publication once it meets all outstanding technical requirements.

Kind regards,

Mingwu Jin, Ph.D.

Academic Editor

PLOS ONE

Additional Editor Comments (optional):

Reviewers' comments:

Reviewer's Responses to Questions

**Comments to the Author**

1. If the authors have adequately addressed your comments raised in a previous round of review and you feel that this manuscript is now acceptable for publication, you may indicate that here to bypass the “Comments to the Author” section, enter your conflict of interest statement in the “Confidential to Editor” section, and submit your "Accept" recommendation.

Reviewer #1: All comments have been addressed

Reviewer #2: All comments have been addressed

2. Is the manuscript technically sound, and do the data support the conclusions?

Reviewer #1: Yes

Reviewer #2: Yes

3. Has the statistical analysis been performed appropriately and rigorously? 

Reviewer #1: I Don't Know

Reviewer #2: Yes

4. Have the authors made all data underlying the findings in their manuscript fully available?

Reviewer #1: Yes

Reviewer #2: Yes

5. Is the manuscript presented in an intelligible fashion and written in standard English?

Reviewer #1: Yes

Reviewer #2: Yes

6. Review Comments to the Author

Reviewer #1: Both my comments are addressed. Thank you.

This is to fill up the required 100 character counts. Sorry.

Reviewer #2: I agree with reviewer #1 that 0.02 mSv excessive effective dose is very little compared to a normal lung screening scan, which is about 1.5 mSv, not to mention the shakiness of the non-threshold linear radiation safety model that ALARA is based on. Underscan is definitely a much bigger concern and a 'safety range' should be allowed for the operators. This DL-based algorithm provided good overall accuracy for overscan determination. In the future, I would love to see an algorithm that can make decisions for at least 3 conditions: underscan, normal range, (maybe a safety range), and severe overscan, not necessarily by DL algorithms alone.

7. PLOS authors have the option to publish the peer review history of their article (what does this mean?). If published, this will include your full peer review and any attached files.

Reviewer #1: No

Reviewer #2: No

---

## [Editor Report · Acceptance letter]

21 Sep 2022

PONE-D-22-05608R2 

Development of Deep Learning-Assisted Overscan Decision Algorithm in Low-dose Chest CT: Application to Lung Cancer Screening in Korean National CT Accreditation Program 

Dear Dr. Chun:

I'm pleased to inform you that your manuscript has been deemed suitable for publication in PLOS ONE. Congratulations! Your manuscript is now with our production department. 

Kind regards, 

on behalf of

Dr. Mingwu Jin 

Academic Editor

PLOS ONE